# A Scalable and Exact Gaussian Process Sampler Via Kernel Packets

## Abstract

In view of the widespread use of Gaussian processes (GPs) in machine learning models, generating random sample paths of GPs is crucial for many machine learning applications. Sampling from a GP essentially requires generating high-dimensional Gaussian random vectors, which is computationally challenging if a direct method, such as the one based on Cholesky decomposition, is implemented. We develop a scalable algorithm to sample random realizations of the prior and the posterior of GP models with Matérn correlation functions. Unlike existing scalable sampling algorithms, the proposed approach draws samples from the theoretical distributions exactly. The algorithm exploits a novel structure called the kernel packets (KP), which gives an exact sparse representation of the dense covariance matrices. The proposed method is applicable for one-dimensional GPs, and multi-dimensional GPs under some conditions such as separable kernels with full grid designs. Via a series of experiments and comparisons with other recent works, we demonstrate the efficiency and accuracy of the proposed method.

## 1 Introduction

Gaussian processes (GPs) have been widely used in statistical and machine learning applications (Rasmussen, 2003; Cressie, 2015; Santner et al., 2003). The relevant areas and topics include regression (O'Hagan, 1978; Bishop et al., 1995; Rasmussen, 2003; MacKay et al., 2003), classification (Kuss et al., 2005; Nickisch & Rasmussen, 2008; Hensman et al., 2015), Bayesian networks (Neal, 2012), optimization (Srinivas et al., 2009), and so on. GP modeling proceeds by imposing a GP as the prior of an underlying continuous function, which provides a flexible nonparametric framework for prediction and inference problems. When the sample size is large, the basic framework for GP regression suffers from the computational challenge of inverting large covariance matrices. A lot of work has been done to address this issue. Recent advances in scalable GP regression include Nyström approximation (Quinonero-Candela & Rasmussen, 2005; Titsias, 2009; Hensman et al., 2013), random Fourier features (Rahimi & Recht, 2007), local approximation (Gramacy & Apley, 2015), structured kernel interpolation (Wilson & Nickisch, 2015), state-space formulation (Grigorievskiy et al., 2017; Nickisch et al., 2018), Vecchia approximation (Katzfuss & Guinness, 2021), sparse representation (Chen et al., 2022; Ding et al., 2021), etc.

In this article, we focus on the sampling of random GP realizations. Such GPs can be either the prior stochastic processes, or the posterior processes in GP regression. Generating random sample paths of the GP prior or the posterior of the GP regression is crucial in machine learning areas such as Bayesian Optimization (Snoek et al., 2012; Frazier, 2018a;b), reinforcement learning (Kuss & Rasmussen, 2003; Engel et al., 2005; Grande et al., 2014), and inverse problems in uncertainty quantification (Murray-Smith & Pearlmutter, 2004; Marzouk & Najm, 2009; Teckentrup, 2020).

To generate the function of a random GP sample, a common practice is to discretize the input space, and the problem becomes the sampling of a high-dimensional multivariate normal vector. Sampling high-dimensional multivariate normal vectors, however, is computationally challenging as well, as we need to factorize the large covariance matrices. Despite the vast literature of the scalable GP regression, the sampling methodologies are still underdeveloped.

Existing scalable sampling algorithms for GPs are scarce. A recent prominent work is done by Wilson et al. (2020). They proposed an efficient sampling approach called decoupled sampling by exploiting Matheron's rule and combining Nyström approximation and random Fourier feature.

They also generalized it to pathwise conditioning (Wilson et al., 2021) based on Matheron's update, which only needs the sampling from GP priors and is a powerful tool for both reasoning about and working with GPs. Motivated by those work, Maddox et al. (2021) extended Matheron's rule for multi-task GPs and applied it to Bayesian optimization; Nguyen et al. (2021) proposed the first use of such bounds to improve Gaussian process posterior sampling. It is worth noting that each of the above methods enforces a certain approximation scheme to facilitate rapid computation, i.e., none of these methods draws random samples from the theoretical Gaussian distributions exactly.

In this paper, we propose algorithms of sampling from GP priors and GP regression posteriors exactly for one-dimension Matérn kernel with half integer smoothness $\nu$, then we extend it to noiseless data in multi-dimension. We introduce the *kernel packet* (KP) (Chen et al., 2022) as a major tool for sampling and reduce the time complexity to $\mathcal{O}(\nu^3 n)$. This produces a linear-time exact sampler if $\nu$ is not too large. Specifically, our work makes the following contributions:

- We propose an *exact* sampling method for Gaussian processes with (product) Matérn correlations on one-dimensional or multi-dimensional grid points. The computational time grows *linearly* in the size of the grid points.
- We propose an exact and scalable sampler for the posterior Gaussian processes based on one-dimensional data or multi-dimensional data on full grid designs.
- We demonstrate the value of the proposed algorithm in the Bayesian optimization and dynamical system problems.

## 2 BACKGROUND

This section covers the related background of the proposed method. Sections 2.1 and 2.2 introduce GPs, GP regression, and the basic method of GP sampling. In section 2.3, we review a newly introduced covariance matrix representation called the *Kernel Packet* (KP) (Chen et al., 2022), which will help expedite the GP sampling.

### 2.1 GPS AND GP REGRESSION

A GP is a stochastic process whose finite-dimensional distributions are multivariate normal. The probability law of a GP is uniquely determined by its mean function $\mu(\cdot)$ and covariance function $K(\cdot, \cdot)$, and we denote this GP as $\mathcal{GP}(\mu(\cdot), K(\cdot, \cdot))$.

Let $f : \mathcal{X} \to \mathbb{R}$ be an unknown function, $\mathcal{X} \subseteq \mathbb{R}^d$. Suppose the training set consists of $n$ Gaussian observations $y_i = f(\mathbf{x}_i) + \epsilon_i$ with noise $\epsilon_i \sim \mathcal{N}(0, \sigma_\epsilon^2)$. In GP regression, we impose the prior $f \sim \mathcal{GP}(\mu(\cdot), K(\cdot, \cdot))$.

Suppose that we have observed $\mathbf{y} = (y_1, \cdots, y_n)^T$ on $n$ distinct points $\mathbf{X} = \{\mathbf{x}_i\}_{i=1}^n$. The posterior of $f$ given the data is also a GP. Specifically, the posterior evaluation at $m$ untried inputs $\mathbf{X}^* = \{\mathbf{x}_i^*\}_{i=1}^m$ follows $\mathbf{f}_* | \mathbf{y} \sim \mathcal{N}(\boldsymbol{\mu}_{*|n}, \mathbf{K}_{*,*|n})$ with (Williams & Rasmussen, 2006):

$$\boldsymbol{\mu}_{*|n} = \boldsymbol{\mu}_* + \mathbf{K}_{*,n}\left[\mathbf{K}_{n,n} + \frac{\sigma_\epsilon^2}{\sigma^2}\mathbf{I}_n\right]^{-1}(\mathbf{y} - \boldsymbol{\mu}_n), \tag{1}$$

$$\mathbf{K}_{*,*|n} = \sigma^2\left(\mathbf{K}_{*,*} - \mathbf{K}_{*,n}\left[\mathbf{K}_{n,n} + \frac{\sigma_\epsilon^2}{\sigma^2}\mathbf{I}_n\right]^{-1}\mathbf{K}_{n,*}\right), \tag{2}$$

where $\sigma^2 > 0$ is the variance of GP, $\mathbf{I}_n$ is a $n \times n$ identity matrix, $\mathbf{K}_{*,n} = K(\mathbf{x}^*, \mathbf{X}) = \left(K(\mathbf{X}, \mathbf{x}^*)\right)^T = \mathbf{K}_{n,*}^T = \left(K(\mathbf{x}^*, \mathbf{x}_1), \cdots, K(\mathbf{x}^*, \mathbf{x}_n)\right)$, $\mathbf{K}_{n,n} = \left[K(\mathbf{x}_i, \mathbf{x}_s)\right]_{i,s=1}^n$, $\mathbf{K}_{*,*} = \left[K(\mathbf{x}_i^*, \mathbf{x}_s^*)\right]_{i,s=1}^m$, $\boldsymbol{\mu}_n = \left(\mu(\mathbf{x}_1), \cdots, \mu(\mathbf{x}_n)\right)^T$ and $\boldsymbol{\mu}_* = \left(\mu(\mathbf{x}_1^*), \cdots, \mu(\mathbf{x}_m^*)\right)^T$.

In this work, we focus on Matérn correlation functions. One-dimensional Matérn correlation functions (Stein, 1999) are defined as

$$K(x, x') = \frac{2^{1-\nu}}{\Gamma(\nu)}\left(\sqrt{2\nu}\frac{|x - x'|}{\omega}\right)^\nu K_\nu\left(\sqrt{2\nu}\frac{|x - x'|}{\omega}\right), \tag{3}$$

for any $x, x' \in \mathbb{R}$, where $\sigma^2 > 0$ is the variance, $\nu > 0$ is the smoothness parameter, $\omega > 0$ is the lengthscale and $K_\nu$ is the modified Bessel function of the second kind. GPs with Matérn correlations

form a rich family with finite smoothness; their sample paths are $\lceil \nu - 1 \rceil$ times differentiable (Santner et al., 2003). By virtue of its flexibility, Matérn family is deemed a popular choice of correlation functions in spatial statistics (Diggle et al., 2003), geostatistics (Curriero, 2006; Pardo-Iguzquiza & Chica-Olmo, 2008), image analysis (Zafari et al., 2020; Okhrin et al., 2020), and other applications.

A common choice of multi-dimensional correlation structure is the "separable" or "product" correlations given by

$$K(\mathbf{x}, \mathbf{x}') = \prod_{j=1}^{d} K_j(x_j, x_j'), \tag{4}$$

for any $\mathbf{x}, \mathbf{x}' \in \mathbb{R}^d$ where $K_j$ is the one-dimensional Matérn correlation function for each $j$. Although the product of Matérn correlations doesn't have the same smoothness properties with multi-dimensional Matérn correlations, the assumption of separability is used extensively in spatio-temporal statistics (Gneiting et al., 2006; Genton, 2007; Constantinou et al., 2017) because it allows for a simple construction of valid space-time parametric models and facilitates the computational procedures for large datasets in inference and parameter estimation.

## 2.2 SAMPLING

The goal is to sample $f(\cdot) \sim \mathcal{GP}(\mu(\cdot), K(\cdot, \cdot))$. To achieve a finite representation, we discretize the input space and consider the function values over a set of grid points $\mathbf{Z} = \{\mathbf{z}_i\}_{i=1}^{p}$, and the objective is to generate samples $\mathbf{f}_p = \left( f(\mathbf{z}_1), \cdots, f(\mathbf{z}_p) \right)^T$ from multivariate normal distribution $\mathcal{N}(\mu(\mathbf{Z}), K(\mathbf{Z}, \mathbf{Z})) = \mathcal{N}(\boldsymbol{\mu}_p, \mathbf{K}_{p,p})$. The standard sampling method of a multivariate normal distribution is as follows: 1) generate a vector of samples $\mathbf{f}_{p,0}$ whose entries are independent and identically distributed normal, 2) employ the Cholesky decomposition (Golub & Van Loan, 2013) to factorize the covariance matrix $\mathbf{K}_{p,p}$ as $\mathbf{C}_p \mathbf{C}_p^T = \mathbf{K}_{p,p}$, 3) generate the output sample $\mathbf{f}_p$ as

$$\mathbf{f}_p \leftarrow \mathbf{C}_p \mathbf{f}_{p,0} + \boldsymbol{\mu}_p. \tag{5}$$

Sampling a posterior GP can be done in a similar manner. Suppose we have observations $\mathbf{y} = \left( y_1, \cdots, y_n \right)^T$ on $n$ distinct points $\mathbf{X} = \{\mathbf{x}_i\}_{i=1}^{n}$, where $y_i = f(\mathbf{x}_i) + \epsilon_i$ with noise $\epsilon_i \sim \mathcal{N}(0, \sigma_\epsilon^2)$ and $f \sim \mathcal{GP}(\mu(\cdot), K(\cdot, \cdot))$. The goal is to generate posterior samples $\mathbf{f}_* | \mathbf{y}$ at $m$ test points $\mathbf{X}^* = \{\mathbf{x}_i^*\}_{i=1}^{m}$. Because the posterior samples $\mathbf{f}_* | \mathbf{y} \sim \mathcal{N}(\boldsymbol{\mu}_{*|n}, \mathbf{K}_{*,*|n})$ are also multivariate normal distributed according to (1) and (2) in Section 2.1, we can do Cholesky decomposition $\mathbf{C}_{*|n} \mathbf{C}_{*|n}^T = \mathbf{K}_{*,*|n}$, and generate GP posterior sample as

$$\mathbf{f}_* | \mathbf{y} \leftarrow \mathbf{C}_{*|n} \mathbf{f}_{n,0} + \boldsymbol{\mu}_{*|n}, \tag{6}$$

where $\mathbf{f}_{n,0} \sim \mathcal{N}(\mathbf{0}, \mathbf{I}_n)$.

## 2.3 KERNEL PACKETS

In this section, we review the theory and methods of kernel packets by Chen et al. (2022). Suppose $K(\cdot, \cdot)$ is a one-dimensional Matérn kernel defined in (3), and one-dimensional input points $\mathbf{X}$ is ordered and distinct. Let $\mathcal{K} = \text{span}\{K(\cdot, x_j)\}_{j=1}^{n}$. Then there exists a collection of linearly independent functions $\{\phi_i\}_{i=1}^{n} \subset \mathcal{K}$, such that each $\phi_i = \sum_{j=1}^{n} A_j^{(i)} K(x, x_j)$ has a compact support. Then covariance matrix $\mathbf{K}_{n,n} = K(\mathbf{X}, \mathbf{X})$ is connected to a sparse matrix $\phi(\mathbf{X})$ in the following way:

$$\mathbf{K}_{n,n} \mathbf{A}_{\mathbf{X}} = \phi(\mathbf{X}), \tag{7}$$

where both $\mathbf{A}_{\mathbf{X}}$ and $\phi(\mathbf{X})$ are *banded matrices*, the $(l, i)^{\text{th}}$ entry of $\phi(\mathbf{X})$ is $\phi_i(x_l)$. The matrix $\mathbf{A}_{\mathbf{X}}$ consists of the coefficients to construct the KPs, and specifically, $A_j^{(i)}$ is the $(j, i)^{\text{th}}$ entry of $\mathbf{A}_{\mathbf{X}}$. In view of the sparse representation and the compact supportedness of $\phi_j$, the bandwidth of $\mathbf{A}_{\mathbf{X}}$ is $(k-1)/2$, the bandwidth of $\phi(\mathbf{X})$ is $(k-3)/2$, $k := 2\nu + 2$, $\nu$ is the smoothness parameter in (3). We defer the detailed algorithm to establish the factorization (7) to Appendix A.1 and the connections to the state-space GP to Appendix A.3. Here we only emphasize that the algorithm to find $\mathbf{A}_{\mathbf{X}}$ and $\phi(\mathbf{X})$ takes only $\mathcal{O}(\nu^3 n)$ operations and $\mathcal{O}(\nu n)$ storage.

Based on (7), we can substitute $\mathbf{A_X}$ and $\phi(\cdot)$ for $\mathbf{K}(\cdot, \mathbf{X})$ in (1), (2) and rewrite the equations as follows:

$$\boldsymbol{\mu}_{*|n} = \boldsymbol{\mu}_* + \phi^T(\mathbf{X}^*)\big[\phi(\mathbf{X}) + \frac{\sigma_\epsilon^2}{\sigma^2}\mathbf{A_X}\big]^{-1}(\mathbf{y} - \boldsymbol{\mu}_n), \tag{8}$$

$$\mathbf{K}_{*,*|n} = \sigma^2\Big(\mathbf{K}_{*,*} - \phi^T(\mathbf{X}^*)\big[\mathbf{A_X}^T\phi(\mathbf{X}) + \frac{\sigma_\epsilon^2}{\sigma^2}\mathbf{A_X}^T\mathbf{A_X}\big]^{-1}\phi(\mathbf{X}^*)\Big), \tag{9}$$

Because $\phi(\mathbf{X})$ and $\mathbf{A_X}$ are both banded matrices, the summations $\phi(\mathbf{X}) + \frac{\sigma_\epsilon^2}{\sigma^2}\mathbf{A_X}$ and $\sigma^2\phi(\mathbf{X}) + \sigma_\epsilon^2\mathbf{A_X}$ are also banded matrices. Therefore, the time complexity of matrix inversion via KPs is only $\mathcal{O}(\nu^3 n)$.

# 3 SAMPLING WITH KERNEL PACKETS

In this section, we present the main algorithms for sampling from Gaussian process priors and posteriors based on the KP technique. We introduce the algorithms of sampling from one-dimensional GP and multi-dimensional GP respectively in sections 3.1 and 3.2.

## 3.1 SAMPLING FROM ONE-DIMENSIONAL GPS

Consider one-dimensional GPs with Matérn correlations for half integer smoothness $\nu$.

**Sampling from the prior distribution** For a set of grid points $\mathbf{Z} = \{\mathbf{z}_i\}_{i=1}^p$, we first compute the sparse matrices $\mathbf{A_Z}$ and $\phi(\mathbf{Z})$ in (7). Now, instead of a direct Cholesky decomposition of the covariance matrix $\mathbf{K}_{p,p} = K(\mathbf{Z}, \mathbf{Z})$, we consider the Cholesky decomposition of the symmetric positive definite matrix $\mathbf{R_Z} := \mathbf{A_Z}^T\phi(\mathbf{Z}) = \mathbf{A_Z}^T\mathbf{K}_{p,p}\mathbf{A_Z}$. This shifting makes a significant difference: $\mathbf{K}_{p,p}$ is a dense matrix, and the Cholesky decomposition takes $\mathcal{O}(n^3)$ time. In contrast, $\mathbf{R_Z}$, as a multiplication of two banded matrices, is also a banded matrix with bandwidth of no more than $2\nu$. It is well known that the Cholesky factor of a banded matrix is also banded, and the computation can be done in $\mathcal{O}(\nu^3 n)$ time. Suppose $\mathbf{Q_Z}\mathbf{Q_Z}^T = \mathbf{R_Z}$, then multiply $(\mathbf{A_Z}^T)^{-1}\mathbf{Q_Z}$ by a vector $\mathbf{f}_{p,0}$, which is standard multivariate normal distributed. It's not hard to show that $(\mathbf{A_Z}^T)^{-1}\mathbf{Q_Z}\mathbf{f}_{p,0} + \boldsymbol{\mu}_p \sim \mathcal{N}(\boldsymbol{\mu}_p, \mathbf{K}_{p,p})$. Hence, we obtain the samples which have same distribution with $\mathbf{f}_p$. In practical calculation, we may first compute the multiplication $\mathbf{Q_Z}\mathbf{f}_{p,0}$ then compute $(\mathbf{A_Z}^T)^{-1}(\mathbf{Q_Z}\mathbf{f}_{p,0})$ to reduce the time complexity since both $\mathbf{A_Z}$ and $\mathbf{Q_Z}$ are banded matrices. The entire algorithm only costs $\mathcal{O}(\nu^3 n)$ time and $\mathcal{O}(\nu n)$ storage. A summary is given in Algorithm 1.

---

**Algorithm 1** Sampling from one-dimensional GP priors.

---

**Input**: Ordered input dataset $\mathbf{Z}$ defined in section 2.1
1: Compute banded matrices $\mathbf{A_Z}$ and $\phi(\mathbf{Z})$ with respect to the input data $\mathbf{Z}$
2: Obtain a banded matrix; $\mathbf{R_Z} := \mathbf{A_Z}^T\phi(\mathbf{Z})$;
3: Apply Cholesky decomposition to $\mathbf{R_Z}$, get the lower triangular matrix $\mathbf{Q_Z}$ satisfying $\mathbf{Q_Z}\mathbf{Q_Z}^T = \mathbf{R_Z}$;
4: Generate samples $\mathbf{f}_{p,0} \sim \mathcal{N}(\mathbf{0}, \mathbf{I}_p)$;
5: Compute $\mathbf{f}_{kp} := (\mathbf{A_Z}^T)^{-1}\mathbf{Q_Z}\mathbf{f}_{p,0} + \boldsymbol{\mu}_p$.
**Output**: $\mathbf{f}_{kp}$

---

**Sampling from one-dimensional GP regression posteriors.** This can be done by combining the KP technique with the *Matheron's rule*. The Matheron's rule was popularized by Journel & Huijbregts (1976) to geostatistics field. Recently Wilson et al. (2020) rediscovered and exploited it to develop a GP posterior sampling approach. The Matheron's rule is stated as follows. Let $\mathbf{a}$ and $\mathbf{b}$ be jointly Gaussian random vectors. Then the random vector $\mathbf{a}$, conditional on $\mathbf{b} = \boldsymbol{\beta}$, is equal in distribution to

$$(\mathbf{a}|\mathbf{b} = \boldsymbol{\beta}) \stackrel{d}{=} \mathbf{a} + \text{Cov}(\mathbf{a}, \mathbf{b})\text{Cov}(\mathbf{b}, \mathbf{b})^{-1}(\boldsymbol{\beta} - \mathbf{b}), \tag{10}$$

where $\text{Cov}(\mathbf{a}, \mathbf{b})$ is the covariance of $(\mathbf{a}, \mathbf{b})$.

By (10), the exact GP posteriors can be sampled by two jointly Gaussian random variables, and we obtain

$$\mathbf{f}_*|\mathbf{y} \stackrel{d}{=} \mathbf{f}_* + \mathbf{K}_{*,n}\big[\mathbf{K}_{n,n} + \frac{\sigma_\epsilon^2}{\sigma^2}\mathbf{I}_n\big]^{-1}(\mathbf{y} - \mathbf{f}_n - \epsilon), \tag{11}$$

where $\mathbf{f}_*$ and $\mathbf{f}_n$ are jointly Gaussian random variables from the prior distribution, noise variates $\epsilon \sim \mathcal{N}(\mathbf{0}, \sigma_\epsilon^2\mathbf{I}_n)$. Clearly, the joint distribution of $(\mathbf{f}_n, \mathbf{f}_*)$ follows the multivariate normal distribution as follows:

$$(\mathbf{f}_n, \mathbf{f}_*) \sim \mathcal{N}\left(\begin{bmatrix}\boldsymbol{\mu}_n\\\boldsymbol{\mu}_*\end{bmatrix}, \begin{bmatrix}\mathbf{K}_{n,n} & \mathbf{K}_{n,*}\\\mathbf{K}_{*,n} & \mathbf{K}_{*,*}\end{bmatrix}\right). \tag{12}$$

We may apply KP to (11) and get the following corollary:

$$\mathbf{f}_*|\mathbf{y} \stackrel{d}{=} \mathbf{f}_* + \boldsymbol{\phi}^T(\mathbf{X}^*)\left[\boldsymbol{\phi}(\mathbf{X}) + \frac{\sigma_\epsilon^2}{\sigma^2}\mathbf{A}_\mathbf{X}\right]^{-1}(\mathbf{y} - \mathbf{f}_n - \epsilon). \tag{13}$$

Given that Algorithm 1 requires distinct and ordered data points, it's reasonable to assume training set $\mathbf{X}$ doesn't coincide with test set $\mathbf{X}^*$. Also, we can rearrange the combined set of the training set $\mathbf{X}$ and test set $\mathbf{X}^*$ to an ordered set and record the index of reordering. Next, we can utilize the Algorithm 1 to draw a reordered vector in $\mathbb{R}^{n+m}$ from the GP prior and obtain jointly samples $\mathbf{f}_n$ and $\mathbf{f}_*$ by recovering the reordered vector to the original sequence. Finally, we plug $\mathbf{f}_n$ and $\mathbf{f}_*$ into formula (13) to calculate the posterior samples. It's obvious the time complexity of this approach is also $\mathcal{O}(\nu^3 n)$ due to the sparsity of the matrices to be Cholesky decomposed and inverted.

## 3.2 SAMPLING FROM MULTI-DIMENSIONAL GPs

For multi-dimensional GPs, we suppose that the points $\mathbf{X}$ are given by a full grid, defined as $\mathbf{X}^{\mathrm{FG}} = \times_{j=1}^d \mathbf{X}^{(j)}$, where each $\mathbf{X}^{(j)}$ is a set of one-dimensional data points, $A \times B$ denotes the *Cartesian product* of sets $A$ and $B$. Based on the separable structure defined in (4) and a full grid $\mathbf{X}^{\mathrm{FG}}$, we can transform the multi-dimensional problem to a one-dimensional problem since the correlation matrix $K(\mathbf{X}^{\mathrm{FG}}, \mathbf{X}^{\mathrm{FG}})$ can be represented by *Kronecker products* (Henderson et al., 1983; Saatçi, 2012; Wilson & Nickisch, 2015) of matrices over each input dimension $j$:

$$K(\mathbf{X}^{\mathrm{FG}}, \mathbf{X}^{\mathrm{FG}}) = \bigotimes_{j=1}^d K_j(\mathbf{X}^{(j)}, \mathbf{X}^{(j)}). \tag{14}$$

**Prior** To sample from GP priors over a full grid design $\mathbf{Z}^{\mathrm{FG}} = \times_{j=1}^d \mathbf{Z}^{(j)}$, it's easy to compute that tensor product $\bigotimes_{j=1}^d \left(\mathbf{A}_{\mathbf{Z}^{(j)}}^{-T}\mathbf{Q}_{\mathbf{Z}^{(j)}}\right)$ is the Cholesky factor of the correlation matrix with a full grid $\mathbf{Z}^{\mathrm{FG}}$. For each dimension $j$, we can generate matrices $\mathbf{A}_{\mathbf{Z}^{(j)}}$, $\boldsymbol{\phi}(\mathbf{Z}^{(j)})$, and Cholesky factor $\mathbf{Q}_{\mathbf{Z}^{(j)}}$ with respect to one-dimensional dataset $\mathbf{Z}^{(j)}$. To get the GP prior samplings over full grids, we need to apply the tensor product $\bigotimes_{j=1}^d \left(\mathbf{A}_{\mathbf{Z}^{(j)}}^{-T}\mathbf{Q}_{\mathbf{Z}^{(j)}}\right)$ to $\mathbf{f}_{p,0}^{\mathrm{FG}}$. More specifically, the samples from GP priors can be computed via

$$\mathbf{f}_{\mathrm{kp}}^{\mathrm{FG}} := \bigotimes_{j=1}^d \left(\mathbf{A}_{\mathbf{Z}^{(j)}}^{-T}\mathbf{Q}_{\mathbf{Z}^{(j)}}\right)\mathbf{f}_{p,0}^{\mathrm{FG}} + \boldsymbol{\mu}_p^{\mathrm{FG}} \sim \mathcal{N}(\boldsymbol{\mu}_p^{\mathrm{FG}}, \mathbf{K}(\mathbf{Z}^{\mathrm{FG}}, \mathbf{Z}^{\mathrm{FG}})). \tag{15}$$

Accordingly, on full grids, we only need to perform a for loop over dimensions, multiply banded matrices in each dimension, and calculate a tensor product. The total method is also $\mathcal{O}(\nu^3 \prod_{j=1}^d n_j)$ because we are also dealing with sparse matrices, here $n_j$ is the number of data points $\mathbf{X}^{(j)}$.

**Posterior** With regard to GP posteriors, it's impossible to employ the prior sampling scheme to draw jointly distributed samples $(\mathbf{f}_n^{\mathrm{FG}}, \mathbf{f}_*^{\mathrm{FG}})$ since we cannot order a sequence of multi-dimensional data. We consider directly Cholesky factorizing the posterior correlation matrix $\mathbf{K}_{*,*|n}^{\mathrm{FG}}$ to obtain $\mathbf{C}_{*|n}^{\mathrm{FG}}$, then use (6) to generate samples. The calculation of posterior mean $\boldsymbol{\mu}_{*|n}^{\mathrm{FG}}$ and posterior variance $\mathbf{K}_{*,*|n}^{\mathrm{FG}}$ only costs $\mathcal{O}(\nu^3 m \prod_{j=1}^d n_j)$ time by using equations (25) and (26) in Appendix A.2. Therefore the time complexity of the entire scheme requires $\mathcal{O}(\nu^3 m^3 + m \prod_{j=1}^d n_j)$.

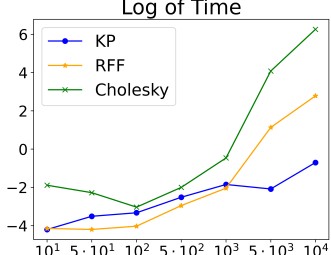 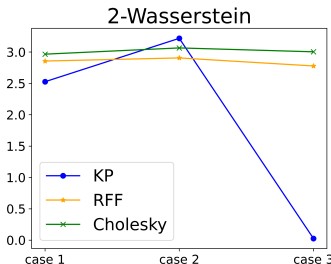

Figure 1: Time and accuracy of different algorithms for sampling from one-dimensional GP priors with Matérn 3/2. We denote our approach (KP) by blue dots, random Fourier features (RFF) by orange stars, direct Cholesky decomposition by green crosses. *Left:* Logarithm of time taken to generate a draw from GP priors, $x$-axis is the number of grid points $p$. *Right:* 2-Wasserstein distances between priors and true distributions over ten points for three different cases $\{z_i\}_{i=1}^{10}$, $\{z_i\}_{i=250}^{259}$, $\{z_i\}_{i=491}^{500}$ when $p = 500$.

## 4 EXPERIMENTS

In this section, we will demonstrate the computational efficiency and accuracy of the proposed sampling algorithm. We first generate samples from GP priors and posteriors with one-dimensional space in section 4.1 and two-dimensional full grid designs in section 4.2. Then we consider the same applications as in (Wilson et al., 2020) and perform our approach to two real problems in section 4.3. For prior samplings, we use the random Fourier features (RFF) with 1024 features (Rahimi & Recht, 2007) and the Cholesky decomposition method as benchmarks. For posterior samplings, decoupled (Wilson et al., 2020) algorithm with exact Matheron's update and the Cholesky decomposition method are used as benchmarks. We consider Matérn correlations in (3) with lengthscale $\omega = \sqrt{2\nu}$, smoothness $\nu = 3/2$ and $5/2$. In two-dimensional problems, we choose "separable" Matérn correlations mentioned in Section 2.1 with the same correlations and same parameters $\omega = \sqrt{2\nu}$ in each dimension. We set the variance as $\sigma^2 = 1$ for all experiments. We set seed value as 99 and perform 1000 replications for each experiment. All plots regarding $\nu = 3/2$ are given in the main article and these associated with $\nu = 5/2$ are deferred to Appendix A.5.

### 4.1 ONE-DIMENSIONAL EXAMPLES

**Prior Sampling** We generate one-dimensional prior samples on uniformly distributed points $\mathbf{Z} = \{z_i\}_{i=1}^p$ over interval $[0, 10]$ with $p = 10, 50, 100, 500, 1000, 5000, 10000$. Left plots in Figure 1 and Figure 7 show the time taken in sampling schemes for different algorithms over the different number of points $p$, we can observe that KP algorithm costs much less than other algorithms for both Matérn 3/2 and Matérn 5/2 correlations especially when $p = 5000, 10000$. Also, the curves of Cholesky decomposition are incomplete due to the limit of storage. To test the accuracy, we select three subsets of size ten: $\{z_i\}_{i=1}^{10}$, $\{z_i\}_{i=250}^{259}$, $\{z_i\}_{i=491}^{500}$ when $p = 500$ and compute the 2-Wasserstein distances between empirical priors and true distributions over these three subsets (called Cases 1-3 thereafter). The 2-Wasserstein distance measures the similarity of two distributions. Let $f_1 \sim \mathcal{N}(\mu_1, \Sigma_1)$ and $f_2 \sim \mathcal{N}(\mu_2, \Sigma_2)$, the 2-Wasserstein distance between the Gaussian distributions $f_1$, $f_2$ on $L^2$ norm is given by (Dowson & Landau, 1982; Gelbrich, 1990; Mallasto & Feragen, 2017)

$$W_2(f_1, f_2) := \left( ||\mu_1 - \mu_2||^2 + \mathrm{tr}\big(\Sigma_1 + \Sigma_2 - 2(\Sigma_1^{\frac{1}{2}} \Sigma_2 \Sigma_1^{\frac{1}{2}})^{\frac{1}{2}}\big) \right)^{\frac{1}{2}}. \tag{16}$$

For the empirical distributions, the parameters are estimated from the replicated samples. From right plots in Figure 1 and Figure 7, we can observe that KP algorithm outperforms RFF and Cholesky method greatly in Case 1 and Case 3, which are boundary points of $\mathbf{Z}$. It may be because of the numerical precision that Cholesky method has the lower 2-Wasserstein distances in Case 3.

**Posterior Sampling** We investigate the performance of sampling over $m = 1000$ test points on the one-dimensional test function, the shifted and dilated Gramacy & Lee function (Gramacy &

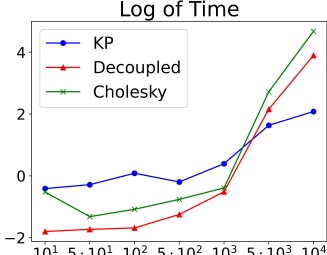 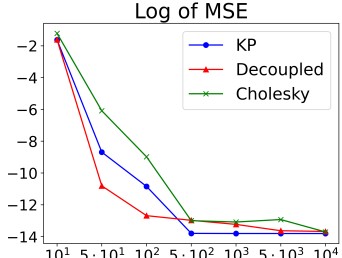

Figure 2: Time and accuracy of different algorithms for sampling from one-dimensional GP posteriors with Matérn 3/2. We denote decoupled method by red triangles. *Left:* Logarithm of time taken to generate a draw from GP posteriors over $m = 1000$ points, $x$-axis is the number of observations $n$. *Right:* Logarithm of MSE over $m = 1000$ points.

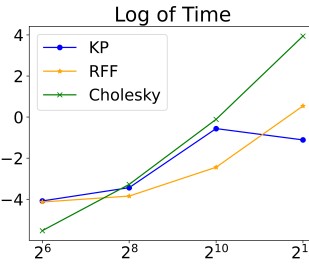 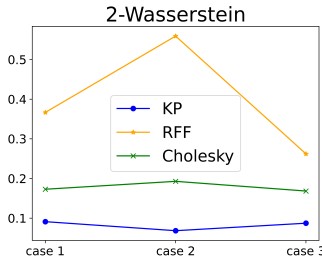

Figure 3: Time and accuracy of different algorithms for sampling from GP priors over full grids with Matérn 3/2. *Left:* Logarithm of time taken to generate a draw from GP priors. *Right:* 2-Wasserstein distances between priors and true distributions over nine points for three different cases $\times_{j=1}^d \{z_i\}_{i=1}^3$, $\times_{j=1}^d \{z_i\}_{i=15}^{17}$, $\times_{j=1}^d \{z_i\}_{i=29}^{31}$ when $\{z_i\}_{i=1}^{31} = \{-5 + 10 \cdot 2^{-5}, \cdots, 5 - 10 \cdot 2^{-5}\}$.

Lee, 2012) $f(x) = \sin(2\pi x + 5\pi)/(0.4x + 1) + (0.2x - 0.5)^4$ over the interval $[0, 10]$ with $n = 10, 50, 100, 500, 1000, 5000, 10000$ respectively given $\sigma_\epsilon = 10^{-3}$. Figure 2 and Figure 8 show the time cost of posterior GP sampling and *Mean Squared Error* (MSE) between the posterior samplings and the true function values with the above settings. It's clear that KP consumes less time to achieve better accuracy compared with decoupled approach and direct Cholesky decomposition method. The performance curves of Cholesky factorization in Figure 2 and Figure 8 in Appendix A.5 are incomplete, because the Cholesky decomposition function fails to work for such a huge sample size and gives a runtime error.

## 4.2 TWO-DIMENSIONAL EXAMPLES

**Prior Sampling** We generate prior samples over level-$\eta$ full grid design: $\mathbf{Z}_\eta^{\mathsf{FG}} = \times_{j=1}^d \{-5 + 10 \cdot 2^{-\eta}, -5 + 2 \cdot 10 \cdot 2^{-\eta}, \ldots, 5 - 10 \cdot 2^{-\eta}\}$ with $\eta = 3, 4, 5, 6$ and $d = 2$. Likewise, we use 2-Wasserstein distances defined in (16) to evaluate the accuracy of prior samplings in three cases, $\times_{j=1}^d \{z_i\}_{i=1}^3$, $\times_{j=1}^d \{z_i\}_{i=15}^{17}$, $\times_{j=1}^d \{z_i\}_{i=29}^{31}$ when $\{z_i\}_{i=1}^{31} = \{-5 + 10 \cdot 2^{-5}, \cdots, 5 - 10 \cdot 2^{-5}\}$. Left plots in Figure 3 and Figure 9 in Appendix A.5 show that KP takes less time than RFF and the Cholesky method especially for $2^{12}$ grid points. Right plots in Figures 3 and Figure 9 show that KP has higher accuracy than RFF for both Matérn 3/2 and 5/2 correlations but has lower accuracy than the Cholesky method for Matérn 5/2 correlations.

**Posterior Sampling** We choose the Griewank function (Griewank, 1981)

$$f(\mathbf{x}) = \sum_{i=1}^d \frac{x_i^2}{4000} + \prod_{i=1}^d \cos(\frac{x_i}{\sqrt{i}}) + 1, \quad \mathbf{x} \in (-5, 5)^d$$

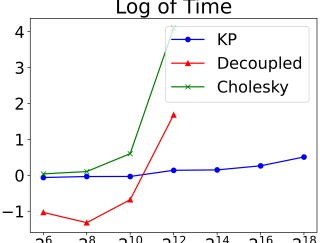 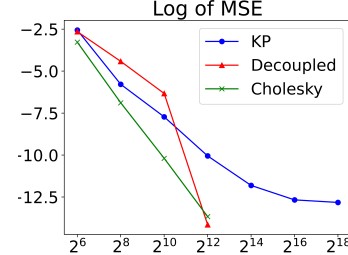

Figure 4: Time and accuracy of different algorithms for sampling from GP posteriors over full grids with Matérn 3/2. *Left:* Logarithm of time taken to generate a draw from GP posteriors over 1024 points. *Right:* Logarithm of MSE over 1024 points.

as our test function and level-$\eta$ full grid design: $\mathbf{X}^{\mathsf{FG}}_\eta = \times_{j=1}^d \{-5+10\cdot 2^{-\eta}, -5+2\cdot 10\cdot 2^{-\eta}, \ldots, 5-10\cdot 2^{-\eta}\}$ with $\eta = 3, 4, \cdots, 9$ and $d = 2$ as our design of experiment. We then investigate the average computational time and MSE over $m = 1024$ random test points for each sampling method. Figure 4 and Figure 10 in Appendix A.5 illustrate the performance of different sampling strategies, we can observe that both direct Cholesky decomposition and decoupled algorithm can only generate posterior samples from at most $2^{12}$ observations due to the limit of storage, however, KP algorithm can sample from $2^{18}$ observations because the space complexity of KP-based computation only requires $\mathcal{O}(\nu n)$. Although the accuracy of the KP method is lower than the direct Cholesky decomposition method and decoupled algorithm, it is still of high accuracy and the time cost is in a much shorter period of time compared with other algorithms.

### 4.3 APPLICATIONS

**Thompson Sampling** Thompson Sampling (TS) (Thompson, 1933) is a classical strategy for decision-making by selecting actions $x \in \mathcal{X}$ that minimize a black-box function $f : \mathcal{X} \to \mathbb{R}$. In round $t$, TS selects $x_{t+1} \in \arg\min_{x \in \mathcal{X}}(f|\mathbf{y})(x)$, $\mathbf{y}$ is the observation set. Upon finding the minimizer, we may obtain $y_{t+1}$ by evaluating at $x_{t+1}$, and then add $(x_{t+1}, y_{t+1})$ to the training set.

In this experiment, we consider a one-dimensional cut of the Ackley function (Ackley, 2012)

$$f(x) = -20\exp\{-0.2 \cdot \sqrt{0.5 \cdot x^2}\} - \exp\{-0.5(\cos(2\pi x) + 1)\} + \exp\{1\} + 20. \tag{17}$$

The goal of this experiment is to find the global minimizer of function in (17). We start with $k = 2\nu + 2$ samples before the optimization, then at each round of TS, we draw a posterior sample $f|\mathbf{y}$ on 1000 uniformly distributed points over the interval $[-5, 5]$ given the observation set. Next, we pick the smallest posterior sample at this round and add it to the training set, and repeat the above process. After some steps, we are able to get closer to the global minimum. In Figure 5, we compare the logarithm of total regret of different sampling algorithms, both the proposed approach (KP) and the decoupled method can find the global minimum within 15 rounds, which outperform the direct Cholesky factorization sampling scheme.

**Simulating Dynamical Systems** Gaussian process posteriors are also commonly used in dynamical systems when we don't have sufficient data. Consider a one-dimensional ordinary differential equation

$$x'(s) = f(s, x(s)), \tag{18}$$

we can discretize the system's equation (18) to a difference equation in the following formula by Euler method (Butcher, 2016),

$$y_t = x_{t+1} - x_t = \tau f(s_t, x_t), \tag{19}$$

where $\tau$ is the fixed step size. We aim to simulate the state trajectories of this dynamical system. First, we set an initial point $x_0$, then at iteration $t$, we draw a posterior GP sampling from the conditional distribution $p(y_t|D_{t-1})$, where $D_{t-1}$ denotes the set of the data $\{(x_i, y_i)\}_{i=1}^n$ and the current trajectory $\{(x_j, y_j)\}_{j=1}^{t-1}$. In our implement, we choose the model as $f(s, x(s)) = 0.5x -$

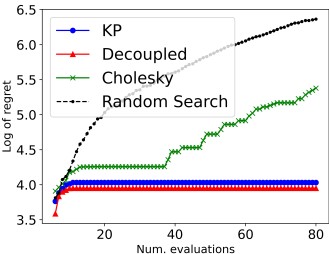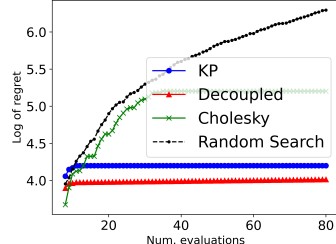

Figure 5: Logarithm of regret of Thompson sampling methods when optimizing Ackley function with Matérn 3/2 (*left*) and Matérn 5/2 (*right*).

$0.05x^3$ with step size $\tau = 0.25$ and initial point $x_0 = -4.5$. The training set was generated by evaluating (19) for $n = 1000$ points $\{x_i\}_{i=1}^n$ uniformly from the interval $[-5, 5]$. Variations $y_t$ in each step were simulated by independent Matérn 3/2 GPs with $\omega = \sqrt{3}$. Figure 6 demonstrates the state trajectories of each algorithm for $T = 480$ steps, time cost in each iteration, and logarithm of MSE between $x_t$ obtained from GP-based simulations and $x_t$ obtained by directly performing the Euler method in (19) at each iteration $t$. The left plot in Figure 6 shows that the KP algorithm can accurately characterize state trajectories of this dynamical system. The middle and right plots in Figure 6 indicate that the KP algorithm takes much less time and yields high accuracy in each step. It achieves nearly the same performance as the decoupled method and outperforms the Cholesky decomposition method.

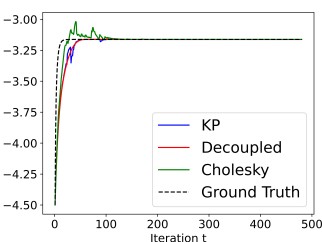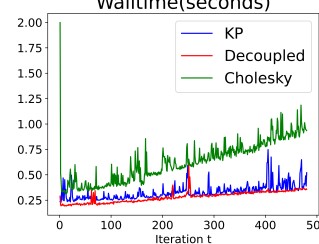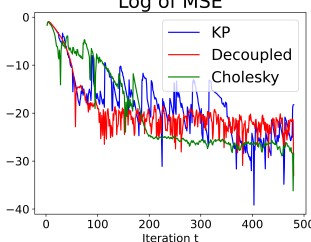

Figure 6: Simulations of an ordinary differential equation. *Left:* Trajectories generated via different algorithms. *Middle:* Time cost at each iteration. *Right:* Logarithm of MSE between simulations and ground truth state trajectories at each iteration.

## 5 DISCUSSION

In this work, we propose a scalable and exact algorithm for one-dimensional Gaussian process sampling with Matérn correlations for half-integer smoothness $\nu$, which only requires $\mathcal{O}(\nu^3 n)$ time and $\mathcal{O}(\nu n)$ space. The proposed method can be extended to some multi-dimensional problems such as noiseless full grid designs by using tensor product techniques. If the design is not grid-based, the proposed algorithm is not applicable, we may use the method in (Ding et al., 2020) to devise approximation algorithms for sampling.

While the proposed method is theoretically exact and scalable algorithm, we observe some numerical stability issues (see Appendix A.4) in our experiments. This explains why sometimes the proposed method is not as accurate as the Cholesky decomposition method. Improvements and extensions of the proposed algorithm to overcome the stability issues and accommodate general multi-dimensional observations will be considered in a future work.

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

# A APPENDIX

## A.1 CONSTRUCTION OF BANDED MATRICES IN (7)

**Intermediate KPs** Let $\mathbf{a} = (a_1, ..., a_k)^T$ be a vector with $a_1 < \cdots < a_k$, then intermediate KPs are defined as

$$\phi_{\mathbf{a}}(x) := \sum_{j=1}^{k} A_j K(x, a_j) \tag{20}$$

and the coefficients $A_j$'s can be obtained by solving the $(k-1) \times k$ linear system

$$\sum_{j=1}^{k} A_j a_j^l \exp\{\delta c a_j\} = 0, \tag{21}$$

with $l = 0, \ldots, (k-3)/2$ and $\delta = \pm 1$.

**One-sided KPs** As before, let $\mathbf{a} = (a_1, ..., a_s)^T$ be a vector with $a_1 < \cdots < a_s$, one-sided KP is given by

$$\phi_{\mathbf{a}}(x) := \sum_{j=1}^{s} A_j K(x, a_j), \tag{22}$$

with $(k+1)/2 \le s \le k-1$. For right-sided KPs, we can get coefficients $A_j$'s by solving

$$\sum_{j=1}^{s} A_j a_j^l \exp\{-c a_j\} = 0, \quad \sum_{j=1}^{s} A_j a_j^r \exp\{c a_j\} = 0, \tag{23}$$

where $l = 0, \ldots, (k-3)/2$ and the second term of equation 23 comprises auxiliary equations for the case $s \ge (k+3)/2$ with $r = 0, \ldots, s - (k+3)/2$. Similar to equation 21, equation 23 is an $(s-1) \times s$ linear system. Left-sided KPs are constructed similarly by solving the following equations:

$$\sum_{j=1}^{s} A_j a_j^l \exp\{c a_j\} = 0, \quad \sum_{j=1}^{s} A_j a_j^r \exp\{-c a_j\} = 0, \tag{24}$$

where $l = 0, \ldots, (k-3)/2$ and the second term comprises auxiliary equations for the case $s \ge (k+3)/2$ with $r = 0, \ldots, s - (k+3)/2$.

**KP Basis** Let $\mathbf{X} = \{x_i\}_{i=1}^{n}$ be the one-dimensional input data satisfying $x_1 < \cdots < x_n$, and $K$ a Matérn correlation function with a half-integer smoothness $\nu$. Suppose $n \ge k := 2\nu + 2$. We can construct the following $n$ functions, as a subset of linear space $\mathcal{K} := \text{span}\{K(\cdot, x_j)\}_{j=1}^{n}$:

1. $\phi_1, \phi_2, \ldots, \phi_{(k-1)/2}$, defined as left-sided KPs $\phi_{(x_1, \ldots, x_{(k+1)/2})}, \phi_{(x_1, \ldots, x_{(k+1)/2+1})}, \ldots, \phi_{(x_1, \ldots, x_{k-1})}$,

2. $\phi_{(k+1)/2}, \phi_{(k+1)/2+1}, \ldots, \phi_{n-(k-1)/2}$, defined as KPs $\phi_{(x_1, \ldots, x_k)}, \phi_{(x_2, \ldots, x_{k+1})}, \ldots, \phi_{(x_{n-k+1}, \ldots, x_n)}$,

3. $\phi_{n-(k-3)/2}, \ldots, \phi_{n-1}, \phi_n$, defined as right-sided KPs $\phi_{(x_{n-k+2}, \ldots, x_n)}, \ldots, \phi_{(x_{n-(k-1)/2-1}, \ldots, x_n)}, \phi_{(x_{n-(k-1)/2}, \ldots, x_n)}$.

Here functions $\{\phi_j\}_{j=1}^{n}$ are linearly independent in $\mathcal{K}$, together with the fact that the dimension of $\mathcal{K}$ is $n$, implies that $\{\phi_j\}_{j=1}^{n}$ forms a basis for $\mathcal{K}$, referred to as the *KP basis*.

In equation 7, the $(l, j)^{\text{th}}$ entry of $\phi(\mathbf{X})$ is $\phi_j(x_l)$. In view of the compact supportedness of $\phi_j$, $\phi(\mathbf{X})$ is a banded matrix with bandwidth $(k-3)/2$:

$$
\phi(\mathbf{X}) = \begin{bmatrix}
\ddots & & & & & & \\
\ddots & \phi_{j-\frac{k-3}{2}}\left(x_{j-2\frac{k-3}{2}}\right) & & & & & \\
\ddots & \vdots & \ddots & & & & \\
 & \phi_{j-\frac{k-3}{2}}(x_j) & \cdots & & \phi_{j+\frac{k-3}{2}}(x_j) & & \\
 & & \ddots & & \vdots & \ddots & \\
 & & & & \phi_{j+\frac{k-3}{2}}\left(x_{j+2\frac{k-3}{2}}\right) & & \ddots \\
 & & & & & & \ddots
\end{bmatrix}.
$$

The matrix of $\mathbf{A_X}$ consists of the coefficients to construct the KPs. $\mathbf{A_X}$ is a banded matrix with bandwidth $(k-1)/2$:

$$
\mathbf{A_X} = \begin{bmatrix}
\ddots & & & & & & \\
\ddots & A_{j-2\frac{k-1}{2},\, j-\frac{k-1}{2}} & & & & & \\
\ddots & \vdots & \ddots & & & & \\
 & A_{j,\, j-\frac{k-1}{2}} & \cdots & & A_{j,\, j+\frac{k-1}{2}} & & \\
 & & \ddots & & \vdots & \ddots & \\
 & & & & A_{j+2\frac{k-1}{2},\, j+\frac{k-1}{2}} & & \ddots \\
 & & & & & & \ddots
\end{bmatrix}.
$$

## A.2 Sampling from Multi-dimensional GP Posteriors

Combine (1) and (2) with the properties of the Kronecker product, the posterior mean and variance for a training set $\{\mathbf{X}^{\text{FG}}, \mathbf{y}^{\text{FG}}\}$ with full grid designs $\mathbf{X}^{\text{FG}} = \times_{j=1}^{d} \mathbf{X}^{(j)}$ can be computed via the following equations:

$$
\boldsymbol{\mu}_{*|n}^{\text{FG}} = \boldsymbol{\mu}_{*}^{\text{FG}} + \mathbf{K}_{*,n} \bigotimes_{j=1}^{d} \left( \mathbf{A}_{\mathbf{X}^{(j)}} \phi(\mathbf{X}^{(j)})^{-1} \right) \left( \mathbf{y}^{\text{FG}} - \boldsymbol{\mu}_{n}^{\text{FG}} \right), \tag{25}
$$

$$
\mathbf{K}_{*,*|n}^{\text{FG}} = \sigma^2 \left( \mathbf{K}_{*,*} - \mathbf{K}_{*,n} \bigotimes_{j=1}^{d} \left( \mathbf{A}_{\mathbf{X}^{(j)}} \phi(\mathbf{X}^{(j)})^{-1} \right) \mathbf{K}_{n,*} \right). \tag{26}
$$

## A.3 Kernel Packets and State-space

Despite kernel packets (Chen et al., 2022) and state-space formulation (Grigorievskiy et al., 2017) can train one-dimensional Gaussian process regression with Matérn correlations exactly in linear time, there is no obvious clue to establish any mathematical connections between these methods. It is worth noting that the state-space approach is a sequential algorithm, which needs significantly more efforts to implement parallel computing; in contrast, the kernel packets approach can be paralleled in a trivial way. Besides, like most other supervised learning algorithms, the kernel packets method is able to separate the learning task into two independent steps: training and prediction, and the computational time for each prediction point is only $\mathcal{O}(1)$ when the inputs are uniformly spaced. However, the state-space formulation cannot make this separation (Grigorievskiy et al., 2017), and users have to either provide the prediction points during the training step, or cost $\mathcal{O}(n)$ time for each prediction point.

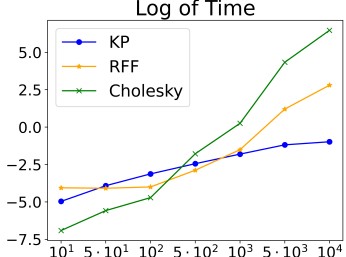 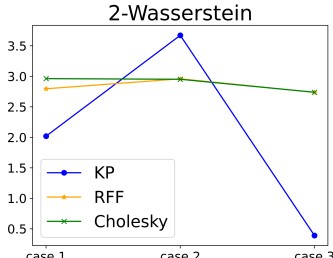

Figure 7: Time and accuracy of different algorithms for sampling from one-dimensional GP priors with Matérn 5/2. *Left:* Logarithm of time taken to generate a draw from GP priors, $x$-axis is the number of grid points $p$. *Right:* 2-Wasserstein distances between priors and true distributions over ten points for three different cases $\{z_i\}_{i=1}^{10}$, $\{z_i\}_{i=250}^{259}$, $\{z_i\}_{i=491}^{500}$ when $p = 500$.

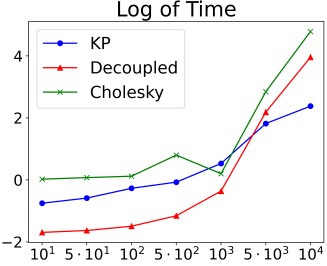 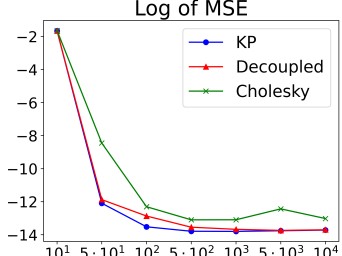

Figure 8: Time and accuracy of different algorithms for sampling from one-dimensional GP posteriors with Matérn 5/2. *Left:* Logarithm of time taken to generate a draw from GP posteriors over $m = 1000$ points, $x$-axis is the number of observations $n$. *Right:* Logarithm of MSE over $m = 1000$ points.

### A.4 NUMERICAL INSTABILITY

In real computation, the matrix $\mathbf{R_Z}$ in Algorithm 1 may not be numerically symmetric positive-definite when the distances between the input points $\mathbf{Z}$ are very small. Since each column of the matrix $\mathbf{A_Z}$ is obtained by solving a null space, each column of $\mathbf{A_Z}$ can be up to a scalar, more work needs to be done to enhance the numerical stability of the algorithm.

### A.5 FIGURES

Figure 7 and Figure 8 show the performance of one-dimensional examples with Matérn 5/2 in section 4.1, Figure 9 and Figure 10 show the performance of two-dimensional examples with Matérn 5/2 in section 4.2.

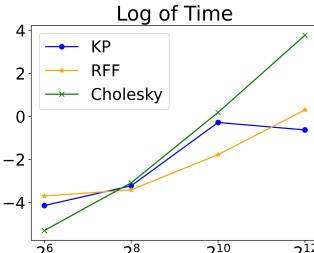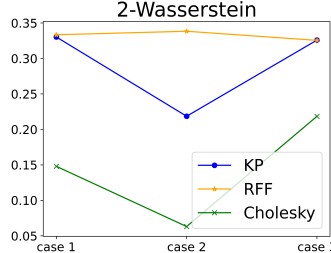

Figure 9: Time and accuracy of different algorithms for sampling from GP priors over full grids with Matérn 5/2. *Left:* Logarithm of time taken to generate a draw from GP priors. *Right:* 2-Wasserstein distances between priors and true distributions over nine points for three different cases $\times_{j=1}^{d} \{z_i\}_{i=1}^{3}$, $\times_{j=1}^{d} \{z_i\}_{i=15}^{17}, \times_{j=1}^{d} \{z_i\}_{i=29}^{31}$ when $\{z_i\}_{i=1}^{31} = \{-5 + 10 \cdot 2^{-5}, \cdots, 5 - 10 \cdot 2^{-5}\}$.

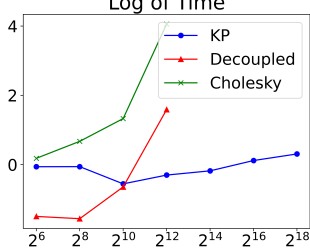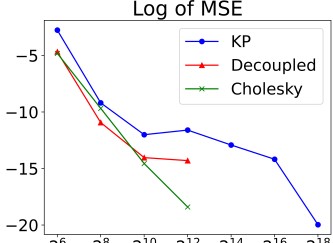

Figure 10: Time and accuracy of different algorithms for sampling from GP posteriors over full grids with Matérn 5/2. *Left:* Logarithm of time taken to generate a draw from GP posteriors over 1024 points. *Right:* Logarithm of MSE over 1024 points.

