# OpenReview forum: "A Scalable and Exact Gaussian Process Sampler via Kernel Packets"
_ICLR.cc/2023/Conference — Submitted to ICLR 2023_

### Official Review · Reviewer_4pJF · 2022-10-16

**Confidence:** 4
**Correctness:** 4
**Technical Novelty And Significance:** 2
**Empirical Novelty And Significance:** 2
**Recommendation:** 3

**Clarity, Quality, Novelty And Reproducibility:**

# Clarity
Overall the paper is reasonably clear; specific suggestions for improving clarity are below:

- Some intuition for why kernel packets exist for Matérn kernels and not other kernels would be useful. Similarly, why does the method only work in one-dimension/require very strict criterion to generalize? I realize this is inherited from prior work on kernel packets (Chen et al, 2022), but because this is so central to the present work still deserves some explanation.
- Include the dimensionality in the big-O statements for the multidimensional case.
- Section 3.1 could be made more clear. In particular, the calculation $A^TKA=A^T\phi(Z)=R$ would useful to write explicitly as this is important to the later algebra, and to justify that $R$ is positive definite so that $Q$ is well defined.

## Typos
- "Gird" $\to$ "Grid", page 2
- 1000 $\to$ 10000 page 6.

# Quality and Reproducibility
- The experiments section is not as thorough as it should be. Several suggestions:
  - Several seeds should be used to indicate the variance in estimates of metrics reported
  - The description of some of the experiments is not clear enough to reproduce. For example, how is the Wasserstein distance computed in equation 16? How many random Fourier features are used in the methods compared to? How is the Matheron's rule update performed in the decoupled method; using Nystrom approximation? If so, what rank/how are the inducing points selected?

# Questions and comments
- Why are the disparities in e.g. MSE so large between KP and Cholesky based implementations? Is this just numerical precision? In exact arithmetic, it seems as though these should be the same. I also wonder how large a role sampling variability plays in these experiments. I don't think estimating sample covariance with samples is likely to be low enough variance to then estimate Wasserstein distance accurately with. How many samples were used? Some indicator of the variance of this estimator would be helpful.
- Similarly, I am a bit surprised to see such a difference between the Cholesky based implementation and other methods in the Thompson sampling example. Is this implemented in float32 or float64? The details of the implementation matter quite a lot for numerical properties (which seems a main point of comparison in experiments), and a detailed description would be useful.
- Why does the RFF method slow down so much for larger sample sizes? It should also scale linearly in sample size (at least for a fixed number of features). Is this due to memory usage?
- The experiments should consider a more detailed comparison of numerical properties of the method. This is often as important as speed of the method.
- I don't see why the Cholesky based implementation was not run for 5000 or 10000 points in figures 1 and 2. I ran this on a laptop in double precision following a similar setup to the one described (using Tensorflow and the GPflow package); I was able to draw a sample in well under a minute and used about 5.6 GBs peak memory. I did run into numerical issues (and had to add a small multiple of the identity to decompose the matrix).
- Why are "the ideas of most scalable GP regression methods (are) not applicable to the sampling problem." (p1_ Both RFFs (as you show in experiments) and Nyström approaches easily extend to sampling.



**Strength And Weaknesses:**

## Strengths
- The proposed idea seems practical.
- Even in one-dimension (where things are inevitably simpler) achieving (near) linear time exact sampling is quite desirable. While the method is quite specific, Matérn kernels are at least widely used as the authors point out.

## Weaknesses
- The contribution beyond work contained in Chen, 2022 is reasonably small. A nice linear-algebraic observation is needed to obtain samples (section 3.1), but I am unsure if this on its own is convincing for a paper without showing the method more convincingly in an interesting application (more discussion of experiments below).
- The scope of the method is narrow. The limitation to Matérn kernels on its own is already quite narrow. This is made more restrictive by the assumption of one-dimensional data. Note also that the multi-dimensional Matérn kernel that is generally used is not a product of one-dimensional  Matérn kernels (this should be made explicit in the text). The product of Matérn kernel considered does not have the same smoothness properties generally associated to Matérn kernels, which seems to be a disadvantage. A bit more explanation in the text as to why the method cannot be generalized to a wider class of kernels would be useful.
- The experiments are not described as thoroughly as they should be, and could certainly be strengthened, more specific comments are below.


**Summary Of The Paper:**

The authors consider the problem of sampling from Gaussian processes defined using the Matérn kernel (both the prior and posterior) in one-dimension. Building on the "kernel packet" approach of Chen et al, 2022, the authors show that samples can be drawn in time that is nearly linear in the number of points sampled. Further, these samples are "exact" in the sense that any error is due to machine precision. Samples from the posterior can be drawn by combining this approach with Matheron's rule, similar to as in Wilson et al, 2020. The approach can be extended to several dimensions if products of one-dimensional Matérn kernels are considered and points are sampled on a grid. The authors provide several small scale experiments to validate the proposed approach.

**Summary Of The Review:**

While narrow in scope, the method seems close to being practical for an interesting task, which is a strong indicator in favor of acceptance. However, I would like to see this practicality demonstrated more convincingly in experiments. The paper falls short in this regard, and doesn't appear to provide sufficient detail to reproduce them (particularly for baseline methods). Crucially, a method must be numerically stable as well as fast to be practical, and I think the paper would be strengthened by some discussion of why there is often such a large discrepancy between the method presented and the Cholesky based method, which is also "exact" ignoring numerical issues.

It is also theoretically interesting that samples can be drawn from these GP priors/posteriors in (nearly) linear time, although I think most of the (theoretical) novelty in this regard comes from Chen et al 2022.

I am currently leaning toward rejection of the paper on balance, but I think it is borderline

---

> ### Author Response · Authors · 2022-11-19
> **Official Response to Reviewer 4pJF**
>
> We appreciate the comments from the reviewer on the manuscript. We made the following revisions based on these suggestions:
>
> 1. In response to the comment “the multi-dimensional Matérn kernel that is generally used is not a product of one-dimensional Matérn kernels (this should be made explicit in the text)”, we added the sentence “the product of Matérn correlations doesn’t have the same smoothness properties with multi-dimensional Matérn correlations” in section 2.1.
>
> 2. In response to the suggestion “Include the dimensionality in the big-O statements for the multidimensional case.”, we incorporated the dimension d in the big-O statements for the multidimensional case in section 3.2
>
> 3. In response to the suggestion “the calculation  ATKA=ATϕ(Z)=R would be useful to write explicitly as this is important to the later algebra”, we added “R = A^T K A=A^T ϕ(Z) ” in section 3.1.
>
> 4. Typos have been corrected
>
> 5. In response to the suggestion “several seeds should be used to indicate the variance in estimates of metrics reported”, we specified seeds used in experiments at the beginning of section 4
>
> 6. In response to the questions “How many random Fourier features are used in the methods compared to? How is the Matheron's rule update performed in the decoupled method; using Nystrom approximation?”, the number of random Fourier features used in the methods and the details of the decoupled method were specified at the beginning of section 4.
>
> 7. In response to the question “Why are the disparities in e.g. MSE so large between KP and Cholesky based implementations?”, we think large MSE between KP and Cholesky based implementations is just because of the numerical precision.
>
> 8. In response to the question “such a difference between the Cholesky based implementation and other methods in the Thompson sampling example”, the Cholesky decomposition was implemented with Numpy float64 in Thompson sampling.
>
> 9. In response to the question “the reason why RFF slows down so much for larger sample sizes”, it may be due to the memory usage of the device we used.
>
> 10. In response to the question “why the Cholesky based implementation was not run for 5000 or 10000”, we added the Cholesky method for 5000 and 10,000 by using the GPflow package.
>
> 11. "the ideas of most scalable GP regression methods (are) not applicable to the sampling problem." was revised to “Despite the vast literature of the scalable GP regression, the sampling methodologies are still underdeveloped” in the Introduction section.

---

> > ### Comment · Reviewer_4pJF · 2022-11-21
> > **Response to rebuttal**
> >
> > I have read the response as well as other viewers comment. I have don't feel my concerns regarding experiments have been adequately addressed. I am particularly concerned about the lack of discussion about numerical precision given large discrepancies in performance of the methods. The sentence added in the appendix seems to me to not address these in a substantial way.
> >
> > As an example of a remaining concern raised earlier that has not been addressed:
> > If I understand correctly, the Wasserstein distance in figure 1 is computed by:
> > 1). Computing the parameters of a Gaussian distribution at e.g. 10 points on a grid.
> > 2.) Sampling from this Gaussian with each of the methods (it isn't totally clear to me how many samples are used for this from the text in the relevant section/caption, perhaps 1000?)
> > 3.) Using these samples, estimate a mean and covariance.
> > 4.) Performing some sort of decomposition (not specified, perhaps SVD?) to compute the square roots needed using the sample matrix and the computed kernel matrix.
> >
> > I would be very skeptical of taking a decomposition of an empirical estimate of a covariance matrix, even if one has access to many samples from a sampler with no numerical error if the matrix is large of ill-conditioned. In particular, I would expect the error from statistical estimation (magnified by taking a decomposition) to be larger than the error from numerical precision/approximation in most instances with a procedure like this. The type of decomposition used will make a big difference, but sampling error, even with many, many samples will likely not behave well under matrix decomposition. This again goes back to my earlier concern, also raised by several other reviewers about the lack of a proper discussion on numerical precision.
> >
> > Even with a more thorough discussion of numerical issues, I think Given that I have remaining concerns regarding the experimental procedure, and after reading concerns raised by other reviewers, particularly regarding limited discussion and comparison to existing approaches, I have lowered my score.

---

### Official Review · Reviewer_xLQn · 2022-10-22

**Confidence:** 4
**Correctness:** 4
**Technical Novelty And Significance:** 2
**Empirical Novelty And Significance:** 2
**Recommendation:** 3

**Clarity, Quality, Novelty And Reproducibility:**

The paper is mostly clearly written and of sufficiently high quality. There are some typos here and there, mostly gird instead of grid. On page 5, in the first paragraph from section 3.2, you say "where each X_j is a set ...", shouldn't j be a superscript, i.e. "where each X^j ..."?

Novelty of the paper is low. In terms of reproducibility, I believe the results to be fairly reproducible. Did not attempt to reproduce the results myself, so I can't be more precise than this.

**Strength And Weaknesses:**

The proposed algorithm is interesting as it enables efficient sampling in the case of Matern correlation functions. The paper can be overall seen as a mix of the results in [Chen et al.] and [Wilson et al.]: kernel packets from [Chen et al.], Matheron's rule from [Wilson et al.]. There's nothing inherently wrong in doing so, however, the novelty factor is low.

A minor weakness is that not enough credit is given to the two source papers. Take for example the numerical applications, why was the Griewank function chosen section 4.2 ? Or why was the Thompson sampling considered in section 4.3 ? There is nothing wrong in considering the same examples as other papers, more so as you compare to them. There is nothing wrong with a sentence the likes of "we consider the same application as in [insert paper citation here]".

A moderate weakness is the lack of interpretation of the numerical results. You content most of the times in just pointing out that the results are better. When it comes for computational cost, there is no issue in interpreting the results. However, when it comes to error analysis, it is not that obvious or not all the implications are obvious. Consider for example the results in figure 1. For case 1 and 3 you have a lower 2-Wasserstein distance than Cholesky decomposition. For me it is not clear why an exact method, your approach, is better than another exact method, Cholesky decomposition approach. Such an explanation would have been more than welcome. You also mention in the conclusion part about some numerical stability issues. It would have been beneficial to mention those issues when the results were presented. It would have been even better to give at least a hint at the cause.

[Chen et al.] - "Kernel Packet: An Exact and Scalable Algorithm for Gaussian Process Regression with Matérn Correlations", Journal of Machine Learning Research 23 (2022) 1-32

[Wilson et al.] - "Efficiently sampling functions from gaussian process posteriors", Proceedings of the 37 th International Conference on Machine Learning, Online, PMLR 119, 2020

**Summary Of The Paper:**

The paper proposes an exact and scalable sampling algorithm for Gaussian processes when using Matern correlation functions. The authors make use of the recent kernel packets formalism that enables a sparse representation of the covariance matrix and reduced computational complexity. Numerical results that show the advantages of the proposed algorithm complete the paper.

**Summary Of The Review:**

The paper proposes a sampling algorithm derived as mix of a recent theoretical discovery, Kernel Packets, and a recently re-discovered rule, Matheron's rule. The novelty factor of the paper is low and there are some issues with the numerical experiments. Leaning towards reject.

---

> ### Author Response · Authors · 2022-11-19
> **Official Response to Reviewer xLQn**
>
> We appreciate the comments from the reviewer on the manuscript. We made the following revisions based on these suggestions:
>
> 1. In response to the comment “not enough credit is given to the two source papers”, we added a sentence "we consider the same application as in [Wilson et al., 2020]" in the experiment section
>
> 2. In response to the question “why an exact method, our approach, is better than another exact method, Cholesky decomposition approach.”  , we think the reason is numerical precision and we added it in the experiment section
>
> 3. In response to the comment “not enough credit is given to the two source papers”, we clarified numerical issues in Appendix A.4
>
> 4. Typos were fixed.

---

> > ### Comment · Reviewer_xLQn · 2022-11-19
> > **Read the rebuttal**
> >
> > I read the rebuttal, however, I'm still not convinced concerning the significance of the contributions.

---

### Official Review · Reviewer_fxzb · 2022-10-26

**Confidence:** 4
**Correctness:** 4
**Technical Novelty And Significance:** 3
**Empirical Novelty And Significance:** Not applicable
**Recommendation:** 6

**Clarity, Quality, Novelty And Reproducibility:**

The paper is clearly written.  This is not a large jump from the previous kernel packets work, but still worthy of investigation.

**Strength And Weaknesses:**

This seems like a reasonable algorithm for fast sampling at large numbers of points, though the restriction to tensor products of 1D Matern kernels (and tensor product grids in the multi-dimensional case) is significant.

The comparisons here are to methods like Random Fourier Features and dense methods.  1D Matern matrices are also very amenable to fast solves with rank-structured matrix techniques (using HSS solvers, HODLR solvers, etc), and these might make a more interesting point of comparison.  Such solvers are available online from Xia, Martinsson, Chandresekaran, and others.

The authors mention a numerical stability issue.  To the extent that they understand the source of this issue, it would be good to clarify what is happening.

**Summary Of The Paper:**

The kernel packet approach allows fast solution for kernel systems with a 1D Matern kernel by reducing linear algebra with the dense kernel matrix to computations with a pair of dense matrices.  On tensor product grids, the tensor product of 1D Matern kernels can be treated the same way.  This approach can be used for fast inference (previous work) or fast sampling (the current paper).

**Summary Of The Review:**

The paper takes the previously-introduced kernel packet decomposition for 1D Matern kernels and applies it not to the problem of training, but rather to the problem of generating sample draws.  The paper is well-written and easy to understand.  Much of the novelty is the previously-introduced kernel packet factorization, but that factorization has not been used in the context of sampling before.  The limitation to tensor products of 1D Matern kernels on tensor product grids is significant, though there are certainly settings where this combination is very natural.

---

> ### Author Response · Authors · 2022-11-19
> **Official Response to Reviewer fxzb**
>
> We appreciate the comments from the reviewer on the manuscript. We made the following revisions based on these suggestions:
>
> 1. In response to the suggestion “The authors mention a numerical stability issue. To the extent that they understand the source of this issue, it would be good to clarify what is happening”, we clarified the numerical stability issues in Appendix A.4.
>
> 2. In response to the suggestion “1D Matern matrices are also very amenable to fast solves with rank-structured matrix techniques (using HSS solvers, HODLR solvers, etc), and these might make a more interesting point of comparison”, we are still working on adding HODLR solver in the experiments and try to do it as much as possible, but may not be able to finish it before the deadline of the rebuttal revision.

---

### Official Review · Reviewer_pohu · 2022-10-28

**Confidence:** 5
**Correctness:** 3
**Technical Novelty And Significance:** 2
**Empirical Novelty And Significance:** 2
**Recommendation:** 3

**Clarity, Quality, Novelty And Reproducibility:**

 - The paper is reasonably well written with a couple of typos e.g. p.2 "gird points", p.3 "In practical calculation", p.6 "p=10,50,100,500,1000,5000,1000", references "gaussian", "wasserstein", "bayesian".
 - The empirical evaluation needs to better reflect the fact that the proposes algorithm is essentially a numerical linear algebra primitive for a structured matrix. So, accuracy should be relative to the exact computations (n=10^4 can be done on a laptop). The timing plots will benefit from a logarithmically scaled ordinate axis.
 - The method goes only mildly beyond the "kernel packet" paper [Chen 22] and hence has only a very limited degree of novelty.
 - The methodology is reasonably simple to be reprogrammed from scratch and the data is mostly synthetic. So the results could be reproduced in principle. However, a well-documented code base would dramatically facilitate reproduction of the results.

**Strength And Weaknesses:**

 + The manuscript is rather well written and simple to understand.
 + The manuscript studies properties of a widely used and well-understood model and provides a scalable sampling algoritm.
 - The method is only applicable to a very small subset of GP models (1D, Matérn covariance, Gaussian regression, rather small smoothness parameter).
 - There is no connection made to the state space representation of GPs. In fact, a scalable algorithm with exactly the same space/time complexity can be obtained via the state space view (see SpInGP https://arxiv.org/abs/1610.08035 and SSGP https://arxiv.org/abs/1802.04846 for two papers where explicit sparse matrices are used). The SpInGP needs to be included as a baseline in the experiments.
 - The manuscript does not make a step forward to better understand the theory behind the "kernel packets" and does not provide insights to allow a statement whether the "kernel packets" are a rediscovery of the state space representation from a different angle or whether the "kernel packets" are something different or more generally applicable beyond the Matérn covariance. Even, numerical comparisons could have given insights.
 - It is well known that SpInGP comes with numerical instabilities. Also the manuscript mentions numerical issues but does not provide a detailed and insightful theoretical or empirical analysis.
 - The numerical experiments do not cover large scale datasets.
 - A code base to replicate the experiments is missing.

**Summary Of The Paper:**

The manuscript proposes a scalable algorithm to draw sample functions from a univariate Gaussian process prior and posterior under the constraint that a Matérn covariance and a Gaussian likelihood is used.
Algorithmically, the method uses the decomposition of the covariance matrix into a product of two banded matrices whose bandwidth depends on the smoothness of the underlying GP. A set of comparative experiments illustrates some properties of the method.

**Summary Of The Review:**

Even though, the "kernel packet" view has not been used for sampling before, the manuscript needs to be improved before getting published. Exact computations and state space computations need to be included in the experiments. The relation between the state space representation needs to be better analysed. Numerical issues are not yet well understood.

---

> ### Author Response · Authors · 2022-11-19
> **Official Response to Review pohu**
>
> We appreciate the comments from the reviewer on the manuscript. We made the following revisions based on these suggestions:
>
> 1. We fixed the typos “gird points”, “p=1000”, the lower cases of the proper nouns such is the default setting of ICLR template
>
> 2. In response to the comment “There is no connection made to the state space representation of GPs The SpInGP needs to be included as a baseline in the experiments.”, we added the clarifications of “kernel packets” and the connections between the “kernel packets” and state-space GP in Appendix A.3. Due to the time limit, we are still working on adding SpInGP as a baseline in the experiments and try to do it as much as possible, but may not be able to finish it before the deadline of the rebuttal revision.
>
> 3. In response to the comment “the manuscript mentions numerical issues but does not provide a detailed and insightful theoretical or empirical analysis”,  we added explanations of numerical issues in Appendix A.4.
>
> 4. In response to the comment “A code base to replicate the experiments is missing”, we added the number of replications we set in the experiments at the beginning of section 4.
> In response to the suggestion “ The timing plots will benefit from a logarithmically scaled coordinate axis”,  we scaled time complexity of the experiment logarithmically.

---

### Decision · Program_Chairs · 2023-01-20

**Decision:**

Reject

**Justification For Why Not Higher Score:**

Clearly below the bar, see reviews

**Justification For Why Not Lower Score:**

N/A

**Metareview: Summary, Strengths And Weaknesses:**

This paper deals with the problem of how to sample from a high-dimensional Gaussian distribution, such as arises for example in Gaussian process models. Drawing joint samples from such models is important in the context of Bayesian optimization, for example to implement Thompson sampling, or more generally to use very general acquisition functions (e.g., as is done in BOTorch).

The proposed method is the combination of two previous papers, and applies only to a rather narrow context, so the novelty is pretty low. Given that, a more thorough and careful evaluation, in particular analyzing numerical robustness, could have carried the work further, but is not given. Also, the authors seem to be missing work along the state space representation, which could be relevant, and certainly should be compared against.